
# Monitoring Sudden Stratospheric Warmings using radio occultation: a new approach demonstrated based on the 2009 event

Ying Li[1], Gottfried Kirchengast[2], Marc Schwärz[2], Florian Ladstädter[2], Yunbin Yuan[1]

[1] State Key Laboratory of Geodesy and Earth's Dynamics, Innovation Academy for Precision Measurement Science and
Technology (APM), Chinese Academy of Sciences, Wuhan, 430071, China
[2] Wegener Center for Climate and Global Change (WEGC) and Institute for Geophysics, Astrophysics, and Meteorology/ Institute of Physics, University of Graz, 8010 Graz, Austria

*Correspondence to*: Ying Li (liying@asch.whigg.ac.cn)

**Abstract.** We introduce a new method to detect and monitor Sudden Stratospheric Warming (SSW) events using Global
Navigation Satellite System (GNSS) Radio Occultation (RO) data at high northern latitudes and demonstrate it for the well-known Jan-Feb 2009 event. We first construct RO temperature, density, and bending angle anomaly profiles and estimate vertical-mean anomalies in selected altitude layers. These mean anomalies are then averaged into a daily-updated $5\,°$ latitude $\times\, 20\,°$ longitude grid over $50\,°$N – $90\,°$N. Based on the gridded mean anomalies, we employ the concept of Threshold Exceedance Areas (TEAs), the geographic areas wherein the anomalies exceed predefined threshold values such as $40\,K$ or
$40\,\%$. We estimate five basic TEAs for selected altitude layers and thresholds and use them to derive primary-, secondary-, and trailing-phase TEA metrics to detect SSWs and to monitor in particular their main-phase (primary- plus secondary-phase) evolution on a daily basis. As an initial setting, the main-phase requires daily TEAs to exceed $3\,\mathrm{Mio.\,km^2}$, based on which main-phase duration, area, and overall event strength are recorded. Using the Jan-Feb 2009 SSW event for demonstration, and employing RO data plus cross-evaluation data from analysis fields of the European Centre for Medium-range Weather
Forecasts (ECMWF), we find the new approach of strong potential for detecting and monitoring SSW events. The TEA metrics show a strong SSW emerging on Jan 17, reaching a maximum on Jan 23, and the strong primary-phase temperature anomaly fading by Jan 27. On Jan 22–23 a MSTA-TEA40 value (TEA of middle stratosphere temperature anomaly $>40\,K$) of about $9\,\mathrm{Mio.\,km^2}$ was reached. The geographic tracking of the SSW showed that it was centered over East Greenland, covering Greenland entirely and extending from Western Norway to Eastern Canada. The secondary- and trailing-phase
metrics track the further SSW development, where the thermodynamic anomaly propagated downward and was fading with a transient upper stratospheric cooling, spanning until end February and beyond. Given the encouraging demonstration results, we expect the method very suitable for long-term monitoring of how SSW characteristics evolve under climate change and variability using both RO and reanalysis data.



## 1 Introduction

Sudden Stratospheric Warming events (SSWs) are strong and highly dynamic phenomena that often occur in the northern polar stratosphere (McInturff et al., 1978; Butler et al., 2015, Butler et al., 2018). Such events are characterized by a rapid increase of temperature (> 30 to 40 K) in the middle and upper stratosphere accompanied by vortex displacements or even

splits (Charlton and Polvani, 2007). Occurrence of SSWs is generally believed to be caused by tropospheric planetary waves which penetrate into the stratosphere, mediated by the Quasi-Biennial Oscillation (QBO) and the Southern Oscillation (SO) in the tropics (Thompson et al., 2002; Labitzke and Kunze, 2009). Such waves influence the stratospheric polar vortex and cause a warming in the upper stratosphere and mesosphere.

The warming will propagate gradually downward and cause an anomalous widespread warming that persists for several

weeks (Baldwin and Dunkerton, 2001; Hitchcock and Shepherd, 2013; Dhaka et al., 2015; Newman et al., 2018). Following the initial warming, a cold anomaly forms in the upper stratosphere that also causes an elevated stratopause (Siskind et al., 2007; Manney et al., 2008; Hitchcock and Shepherd, 2013). The tropical atmosphere is as well found to be influenced (Kodera et al., 2011; Yoshida and Yamazaki, 2011; Dhaka et al., 2015). Cooling can be observed in the tropical stratosphere and also the tropopause is found altered (Yoshida and Yamazaki, 2011; Dhaka et al., 2015). Furthermore, gravity wave

activity, cirrus cloud formation and electron density of ionosphere are all found affected by SSWs (Eguchi, N., Kodera, K. 2010; Yue et al., 2010; Sathishkumar and Sridharan, 2011; Kohma and Sato, 2014). Due to such strong impacts and far-reaching teleconnections of SSWs, it is hence important to detect and monitor SSW events in a robust and reliable way.

The observation and detection of SSWs requires evenly distributed and accurate height-resolved observations of the stratosphere at high latitudes. However, robust techniques providing high-quality observations in these remote regions are

notoriously sparse. Past researches mainly used radiosonde, rocketsonde, conventional satellite or reanalysis data to study SSWs (McInturff et al., 1978; Charlton and Polvani, 2007; Manney et al., 2008, 2009; Hitchcock and Shepherd, 2013). However, both radiosonde and rocketsonde cannot provide evenly-distributed observations due to their mostly land-limited properties. Furthermore, since vulnerable to radiation biases and constrained by elevation limits, few radiosondes can provide data above 30 km (Butler et al., 2015).

With the advent of the satellite era, it became possible to put passive sounding instruments, such as microwave limb sounders and infrared radiometers, on satellites to observe the atmosphere (e.g., Charlton et al., 2007; Manney et al., 2009). Due to the movements of the satellites, observations are globally evenly distributed, in principle. However, satellite passive sounding data come in the form of radiances and no unique solution then exists, in terms of the radiative transfer equation, to accurately convert radiances to height-resolved temperature or winds, which are key variables for SSW monitoring.

Therefore, the fit-for-purpose of measurements from these instruments is limited.

With the development of atmospheric data assimilation systems, re-analysis data have become a quite reliable data source for long-term atmospheric analysis, due to their advantages of regularly distributed data in space and time and their capability to provide data up into the mesosphere (Charlton et al., 2007; Yoshida and Yamazaki, 2011; Butler et al., 2018). However, re-



analysis data may have inhomogeneities and irregularities in the long-term, due to observation system updates and varying analysis biases in sparsely observed domains, which may limit their long-term stability in monitoring SSWs and possible changes in their characteristics due to climate change and interannual variability.

As a consequence of the limitations of classical observations and re-analyses data, there is currently no standard definition of
SSWs. Early definitions were usually based on temperature increases and wind reversals. An often used early definition was provided by McInturff in 1978, presented in one of the reports of World Meteorological Organization (WMO) Commission for Atmospheric Sciences (CAS): 1. A stratospheric warming can be called minor if a significant temperature increase is observed of at least 25 ° degrees in a week or less at any stratospheric level in any area of the wintertime hemisphere and if criteria for major warmings are not met; 2. A stratospheric warming can be said to be major if at 10 mb or below the
latitudinal mean temperature increase poleward from 60 ° degrees and an associated circulation reversal is observed. This definition has been dominated over the 1980s and 1990s though the detailed interpretations could be different, e.g., using observations below 10 mb, or using wind observations at 65 °N degrees, etc.

With the development of observation techniques, several new definitions for characterizing SSWs have been proposed. Butler et al., 2015 made a detailed literature review on the definitions of SSW and discussed as many as 9 often used
definitions of SSWs, such as zonal-mean zonal winds at 10 hPa and 60 ° latitude (Christiansen 2001; Charlton and Polvani, 2007), polar cap-averaged geopotential height anomalies at 10 hPa (e.g., Thompson  et al., 2002), Empirical Orthogonal Functions (EOFs) of gridded pressure-level data of geopotential height anomalies (Baldwin and Dunkerton 2001; Baldwin 2001), zonal wind anomalies (Limpasuvan et al. 2004) or temperature anomalies (e.g., Kuroda and Kodera 2004; Hitchcock and Shepherd 2013; Hitchcock et al., 2013). Each definition has unique characteristics and application purposes, e.g., EOFs
of height anomalies focus more on the stratosphere-troposphere coupling.

One of the most commonly used SSW definitions in recent studies is the one based on zonal-mean zonal wind at 60 °N. This definition has been used in several previous studies though interpretation could be slightly different (e.g., Andrews et al., 1985; Labitzke and Naujokat 2000) and was described in detail by Charlton and Polvani, 2007 (denoted as CP07 below). According to the CP07 definition, a major midwinter warming occurs when the zonal mean zonal winds at 60 °N and 10 hPa
become easterly during winter, defined here as (November-March (NDJFM)). The first day on which the daily mean zonal mean zonal wind at 60 °N and 10 hPa becomes easterly is defined as the central date of the warming. Once SSW events have been identified, they are classified into polar vortex displacements or split ones by identifying the number and relative sizes of cyclonic vortices during the evolution of the warming.

From the above, we can find that it would be impossible to find a single definition to serve every purpose to describe every
event perfectly. However, it is still important to find a standard definition for the purposes of statistical assessments, based on historical data and future climate simulations. Butler et al. (2015) suggest that with the development of observation techniques, it is time again to propose a standard definition of SSWs. The new definition should be proposed primarily for the purpose of describing polar winter variability. Secondly, it should be easily calculated and applicable to reanalysis and



model outputs, both in post-processing and in real time. Finally, the new definition should not be highly sensitive to details, such as an exact latitude, background climatology, threshold wind speed, spatial extent, or pressure level.

Since the early 2000s, Global Navigation Satellite System (GNSS) radio occultation (RO) has become a new and reliable data source for weather and climate studies (e.g., Kursinski et al., 1997; Steiner et al., 2001; Hajj et al., 2002; Anthes, 2011;

Steiner et al., 2011). The RO technique uses GNSS receiver instruments on low Earth orbit satellites to receive GNSS signals for active atmospheric limb sounding in occultation geometry. As the signals propagate through the atmosphere, they are phase-delayed and bent in their path, due to vertical refractivity gradients determined by density and temperature changes. Building on these properties, accurate bending angle profiles can be retrieved from RO signal phase delays, which are highly stable during the measurement time of vertically scanning from mesopause into the troposphere (setting events) or from

troposphere into mesopause (rising events) of just about one minute, called an RO event. The bending angle profile is then converted to a refractivity profile (via an Abel transform), which is directly proportional to the density profile in the stratosphere (refractivity equation), from which then the pressure profile (via hydrostatic integration) and finally temperature profile (via equation of state) is derived.

The vertical resolution of RO in the stratosphere is about 1 km, supporting height-resolved studies, and validation results

against radiosonde and (re-)analysis data suggest that RO data are of small discrepancy to these in the upper troposphere and lower stratosphere (Scherllin-Pirscher et al., 2011a; 2011b; Ladstädter et al., 2015). Finally, RO data can be combined without the need of inter-calibration, which makes them very suitable for climate-related studies (Foelsche et al. 2011; Steiner et al., 2011; 2013; 2020). Due to these distinctive advantages, RO data have been successfully used in many weather and climate studies and are hence a promising data source also for detecting and monitoring SSWs. Since continuous multi-

satellite RO data started in 2006 (see Sect. 2 below), the geographic data coverage is sufficiently dense for monitoring and analyzing regional-scale phenomena such as SSWs. Complementary to reanalysis datasets, which also offer dense coverage, RO reprocessing datasets hence feature an accurate and long-term stable observational data record of climate benchmark quality (Steiner et al., 2020), allowing stable conditions for SSW monitoring over decades. Therefore, given the high complementarity of these observations to reanalysis (Bosilovich et al., 2013; Parker, 2016; Simmons et al., 2020), RO data

well fulfill the requirements presented by Butler et al. (2015).

A couple of studies have used RO data to analyze SSW already. For example, Wang et al. (2009) have used RO to study SSW influences on gravity waves during events in 2007-2008. Yue et al. (2010) and Lin et al. (2012) have used RO data to study ionospheric variations related to the 2009 SSW event. Klingler (2014) has used RO data to examine the temperature changes during the 2009 SSW event, and compared the results to European Centre for Medium-Range Weather Forecasts

(ECMWF) data, while Dhaka et al. (2015) have used them to study the dynamical coupling between polar and tropical regions during this event.

In this study, we use RO data to introduce a new method to detect and monitor SSW events. As a demonstration case, the Jan-Feb 2009 SSW event was used, since this is well known from other studies (such as the ones just cited above) and therefore context knowledge is good. As a cross-check and for evaluation of robustness, ECMWF analysis data are also used



and the results are compared to those with RO data. The paper is arranged as follows. Section 2 introduces the data and methodology. Section 3 introduces the detection and monitoring results. Section 4 provides our conclusions.

## 2 Data and methodology

### 2.1 Radio occultation data

Continuous RO data started in 2001 with the Challenging Mini-satellite Payload mission (CHAMP; Wickert et al., 2001), followed by the Gravity Recovery and Climate Experiment (GRACE; Wickert et al., 2005), the Constellation Observing System for Meteorology, Ionosphere and Climate (COSMIC; Schreiner et al., 2007), the European Meteorological Operational satellites (MetOp; Luntama et al., 2008), the Chinese FengYun-3C operational satellite (Sun et al., 2018), and others. These missions, especially the launch of the COSMIC mission in 2006, which was a constellation of six satellites,

have ensured as of 2006 a sufficient coverage with RO event observations for regional-scale studies such as of SSWs.

In this study, we use the atmospheric RO profile data from the Wegener Center for Climate and Global Change (WEGC), processed by its latest Occultation Processing System version 5.6 (denoted as OPSv5.6 hereafter). Several studies that introduced, validated and evaluated these OPSv5.6 data (e.g., Ladstädter et al., 2015; Schwärz et al., 2016; Angerer et al., 2017; Scherllin-Pirscher et al., 2017) as well as inter-comparison to other RO center datasets (Steiner et al., 2020) show that

the OPSv5.6 stratospheric profiling data of interest in this study are of high quality for the purpose. For a detailed discussion of quality aspects of the OPSv5.6 data we refer to Angerer et al. (2017). We use the high quality-flagged temperature, density, and bending angle profiles over Jan-Feb 2009, the time period of our demonstration study, in the northern high latitude study domain of 50–90 °N.

Figure 1 illustrates the distribution of RO events on 23 Jan 2009 and the number of RO events we used per day over Jan–Feb

2009. The upper panel shows that RO observations are evenly distributed in the study domain from 50 °N to the North Pole, within which strong warmings were found by previous studies of the SSW event (Labitzke and Kunze, 2009; Harada et al., 2010; Kodera et al., 2011; Taguchi et al., 2011). Such a regular distribution applies also to the other days of the study period. The bottom panel shows that daily numbers of RO events within the three successively smaller polar cap regions 50 °– 90 °N, 60 °– 90 °N, and 70 °– 90 °N are within about 500–700, 300–400, and 150–200 RO events per day, respectively. This

is typical for the RO observation period as of 2006 and sufficiently dense for robust SSW monitoring as we will see.

### 2.2 ECMWF analysis data

As mentioned in Sect. 1, a robust SSW definition should not only be applied to observation data, but also be readily applicable to (re)analysis and model outputs with their regular-gridded datasets. Therefore, we also use operational analysis data from the ECMWF over the same study period for cross-check and demonstration of the applicability of our new

approach also to such gridded datasets. The ECMWF analysis fields used are based on T42L91 resolution (sampled at 2.5 °





latitude $\times 2.5\,°$ longitude grids, and 91 hybrid-pressure vertical levels up to about 80 km), and at the four time layers 00, 06, 12, and 18 UTC each day. This corresponds to roughly 300 km horizontal resolution that is similar to RO in the stratosphere (e.g., Kursinski et al., 1997). The 91 vertical levels correspond to about 1 km resolution in the tropopause region and gradually coarser resolution across the stratosphere, up to several kilometers in the mesosphere (Untch et al., 2006).

ECMWF data are used for cross-check in two variants. The first variant is to use the RO-collocated analysis profiles, extracted by interpolation from the analysis fields to the RO event locations, together with the OPSv5.6 RO profiles. We apply the approach in the same way to these collocated analysis profiles as to the RO profiles. We note that while the density and temperature profiles derive directly from analysis field interpolations, the bending angle profiles are obtained from forward modeling (Abelian transform from refractivity profiles) in the OPSv5.6 system.

The second variant is that we directly use the ECMWF analysis data at their regularly gridded resolution of $2.5\,° \times 2.5\,°$, and with 4 time layers per day, which makes the averaging into coarser bins straightforward in this case and hence enables to clearly assess possible (under-)sampling biases if brought in by the limited RO events coverage, given that we intend a monitoring on a daily basis. At the same time this prepares the use of the new method with reanalysis data, such as the new European Reanalysis ERA5 (Hersbach et al., 2019, 2020; Simmons et al., 2020), foreseen in parallel to the use with RO data

in future long-term application over the recent decades.

### 2.3 SSW detection and monitoring method

Table 1 illustrates the methodology of our SSW detection. The first step is to generate RO temperature, density, and bending angle anomaly profiles by using individual RO profiles minus collocated climatological profiles, with the latter extracted from long-term gridded RO climatology fields interpolated to RO locations as described in (1), (2), and (3) of Table 1.

Anomalies of various atmospheric parameters have been successfully used in lots of researches for SSW detection, cloud-top altitude detection and atmospheric blocking (Hitchcock and Shepherd, 2013; Biondi et al., 2015, 2017; Brunner et al., 2016). The long-term climatology was constructed monthly using RO data of the same months over 2007 to 2017. It is based on a $2.5\,°$ latitude $\times 2.5\,°$ longitude grid. At each of the grid centers, RO profiles within 300 km of the same month over the 11 years' period are used for averaging. Sensitivity tests show that our constructed RO climatology show only small differences

to climatologies calculated using ECMWF analysis data.

Based on the climatology, for our time period used as a January-February average, collocated climatological profiles can be obtained through a vertical and horizontal interpolation. Figure 2 shows RO profiles and their anomaly profiles of two exemplary RO events as indicated in Fig. 1. Left panel shows that RO profiles of event1, which locates in the most warming area, deviate more from climatological profiles than that of event2 locating in less warming area. Anomaly profiles shown in

the right panel illustrate consistent larger anomalies of event1.

The next steps are to generate five basic daily updated Thresholds Exceedance Areas (TEAs) as described in (4) – (8) of Table 1. TEA is the the geographic area wherein RO gridded mean anomalies of the day exceed predefined thresholds such as 40 K or 40 %. The first step of calculating TEA is to calculate vertical mean anomaly values of selected stratospheric





altitude ranges. The vertical mean anomalies are then averaged into geographic bins on a $5\,°$ latitude $\times\,20\,°$ longitude grid over the observation area $50 - 90\,°N$, with grid points on latitude circles from $50\,°N$ to $85\,°N$ and on longitude meridians from $10\,°E$ to $350\,°E$ ($8 \times 18$ grid points in total).

In order to allow more RO events coming in for a reliable statistical averaging, we use overlapping bin areas on the $5\,° \times 20\,°$
grid as well as include time-wise, with lower weight, also the neighbor days of the given day. The latitudinal extent of the bins is set to be $10\,°$ ($+/–5\,°$ about grid point latitude) for all latitude circles. Longitudinal bin extents $\Delta\lambda$ are determined to be $30\,°$ ($+/–15\,°$ about grid point longitude) at the $50\,°N$ grid line and then gradually expand with increasing latitude in line with meridian convergence as $\Delta\lambda_\varphi = \Delta\lambda_{50°} \frac{\cos(50°)}{\cos\varphi}$, where $\varphi$ denotes the grid point latitudes from $50\,°N$ to $80\,°N$. At the final $85\,°N$ latitude circle (representing the polar cap area $80 - 90\,°N$), we just directly average data from all longitudes. The
temporal extent is set to be 3 days ($+/–1$ day about given day), with the data of the two neighbor days getting a weight of 0.25 only, while those of the given day are weighted by 0.5. Detailed sensitivity tests showed that these selections of gridding and of spatial and temporal extents are reasonable and robust. Based on this averaging scheme, the number of RO profiles available per grid bin for the daily-updated averaging ranges from 60 to more than 120 profiles.

To examine various atmospheric layers, five basic TEAs are calculated, i.e., MSTA-TEA (Middle Stratosphere Temperature
Anomaly TEA); LMBA-TEA, (Lower Mesosphere Bending angle Anomaly TEA); LSTA-TEA (Lower Stratosphere Temperature Anomaly TEA); USDA-TEA (middle and Upper Stratosphere Density Anomaly TEA); USTA-TEA (Upper Stratosphere Temperature Anomaly TEA). The altitude ranges for calculating these TEAs are selected according to the response altitude ranges of the three anomalies and also the utilities of the TEAs in formulating the metrics. Response altitude ranges are regarded as the altitude ranges where anomalies show distinct increases and decreases to reflect with good
sensitivity the thermodynamic changes caused by an SSW event. Our inspections of small ensembles of individual RO anomaly profiles and also results of Sect. 3.1 on polar mean anomaly profiles suggest that good response altitude ranges of temperature, density and bending angle to SSW are 20–25 km, 30–35 km, 40–45 km, and 50–55 km, respectively.

Based on the chosen response altitude ranges, the variables and ranges actually used for calculating the five TEAs are carefully selected according to their utilities in measuring SSW. MSTA-TEA and LMBA-TEA are used to capture the
sudden warming and are therefore calculated using temperature and bending angle anomalies. LSTA-TEA and USDA-TEA are used to examine the downward propagated warming and therefore they are calculated using temperature and density anomalies in lower response altitude ranges, i.e., 20–25 km for LSTA-TEA and 40–45 km for USDA-TEA. Finally, USTA-TEA is to capture the upper stratospheric cooling in the SSW trailing phase and is calculated using temperature anomalies of 40–45 km.

As thresholds for calculating these five TEAs, we use those defined in Table 1, (4)–(8). The selection of these thresholds was determined by careful sensitivity tests and guided by the results on polar-mean and regional mean anomalies shown in Sects 3.1 and 3.2. Figure 3 illustrates our selection of height ranges of anomaly profiles for calculating the five TEAs based on representative example profiles. The short vertical lines represent vertical mean values in corresponding altitude ranges. For



this RO event, temperature vertical mean anomalies at the 40–45 km, 30–35 km, and 20–25 km ranges are about 25 K, 60 K, and 20 K, respectively. The density vertical mean anomaly at 40–45 km is near 50 % and the bending angle vertical mean anomaly in 50–55 km near 70 %.

Based on the five TEAs, we formulated our SSW metrics as defined in Table 1, (9)–(13), where (9)–(11) are the preferred

metrics while (12)–(13) are fallback metrics for (9)–(10) requiring only temperature as variable. First is the SSW Primary-Phase metric SSW-PP-TEA (9), used to express the main and primary sudden stratospheric warming anomaly strength. It is calculated by averaging the exceedance areas MSTA-TEA>40 K and LMBA-TEA>40%. The Secondary-Phase metric SSW-SP-TEA (10) is used to express the downward propagated warming anomaly strength, and is estimated by averaging the areas LSTA-TEA>25K and USDA-TEA>40%. The Trailing-Phase metric SSW-TP-TEA (11) is expressing the trailing

upper stratospheric cooling anomaly strength, and is estimated by using the area USTA-TEA<–40K.

The preferred primary- and secondary-phase metrics (9) and (10) are constructed as a two-variable estimate (combining temperature and bending angle/density TEAs), since we find them more robust for characterizing the main phase of the SSW then single-variable metrics. However, users who prefer a simplified approach, or who only have stratospheric temperature profiles or fields available (within 20 to 45 km), can use the temperature-only metrics (12)–(13) instead, which do not

include the averaging with the TEAs co-estimated from bending angle (9) or density (10).

Based on the three metrics, either (9)–(11) or (12)–(13) and (11), we can finally detect a SSW event and monitor the strength of the event. We introduce three SSW indicators for this purpose as defined in Table 1, (14)–(16). The first is main-phase duration, SSW-MPD, which indicates the duration of the SSW warming anomaly based on the primary- and secondary-phase metrics. This indicator is estimated by counting the number of days with either the SSW-PP-TEA or the SSW-SP-TEA being

larger than a minimum exceedance area $TEA_{Min}$. The latter is set to the plausible value of 3 Mio. km$^2$ in this demonstration study (an area of ~1000 km effective radius around center location) and may become somewhat adjusted in longer-term application. The second indicator is main-phase area, SSW-MPA, which represents the mean daily threshold exceedance area during the main-phase duration. Combining these two indicators into an area-duration product yields the main-phase strength, SSW-MPS, as the third and overall indicator of the severity of the SSW, enabling a classification into weak,

medium, and strong events for example.

In a follow-on work using long-term RO and reanalysis datasets, these indicators will be used to detect SSW events, for example by requiring a minimum main-phase duration of 7 days or so to qualify as an SSW, and to record the strength of the events. However, the specific settings for robust SSW detection, monitoring, and classification based on the defined duration and area indicators can only be given as part of the application of the new approach to the long-term data.

Below we demonstrate the utility to do so, both for profile-based RO and gridded analysis data, for the Jan-Feb 2009 SSW event. In addition to demonstrating the detection and monitoring approach, we also demonstrate the parallel possibility intrinsic in our TEA-based approach to dynamically track the geographic movements of any event of interest. For this purpose we introduce the parameters Anomaly Maximum/Minimum (AM) value, and the location of these AM values,



which can be used to locate the warming/cooling centers and their geographic track for the five basic TEAs. For convenience, Table 1, (17)–(18), lists and briefly explains also these auxiliary parameters.

## 3 Results and discussion

Section 3.1 presents temporal evolution of polar-cap mean RO anomaly profiles to have a general understanding of the characteristics of RO anomalies. Section 3.2 shows the distribution of RO gridded mean anomalies on several selected days for providing insight on the basic space-time dynamics tracked by the approach. Section 3.3 introduces our detection results of the Jan-Feb 2009 SSW demonstration event in terms of the five basic TEAs at selected thresholds and also discusses the SSW metrics of the event.

### 3.1 Polar-cap mean anomalies

Figure 4 shows the temporal evolution of polar cap ($60\,° – 90\,°N$) mean temperature, density and bending angle anomaly profiles of RO data and collocated ECMWF data during the days of January and February 2009. RO temperature, density, and bending angle anomalies in their response altitude ranges show clear positive anomalies ($>10\,K/>10\,\%$) from Jan 18 that quickly increase up to more than $30\,K/30\,\%$ on Jan 23 and then quickly decrease. Such rapid increase and decrease of positive anomalies indicate a strong and rapid warming in the stratosphere. The positive anomalies propagate downwards to lower altitude levels (middle stratosphere for density and bending angle, lower stratosphere for temperature) and cause longer-lasting anomalous conditions there till the end of February.

Before the sudden and rapid warming, negative anomalies are found for all the three parameters, indicating a moderate precursor cooling of the stratosphere. The cooling signal is imprinted more strongly in the density and bending angle anomalies in the upper stratosphere and lower mesosphere than it is observed in temperature over the lower and middle stratosphere.

After the sudden warming, negative (cooling) anomalies are again found at higher altitude levels than the main response altitude levels, with a particular strong imprint in upper stratosphere temperature, where its fingerprint lasts over many weeks while the altitude of maximum cooling exhibits a slow downward propagation. Related to the chosen altitude layers for computing the five TEAs, we can see that they are defined so that they can well capture the SSW evolution from the initial phase to the trailing phase.

Comparing the RO profiles-based anomalies with the ECMWF analysis-based anomalies, we find that both the magnitudes and dynamical variations of the anomalies from the two datasets are generally consistent below about $50\,km$. The differences are found above $50\,km$, where RO data show larger density and bending angles anomalies and smaller temperature anomalies compared to ECMWF data. These increased differences are attributable to both datasets for the following reasons: (1) ECMWF data are of sparse vertical resolution and with limited constraint from assimilated data above $50\,km$ (e.g., Untch et al., 2006; Simmons et al., 2020), degrading their accuracy; (2) RO data accuracy reaches somewhat higher in bending angle and density profiles (errors $<1\,\%$ to about $50-60\,km$) and less high in temperature ($<1\,\%$ to about $40\,km$); e.g., Steiner et al.,





2020), so that also for these data the accuracy degrades above 50 km. In the follow-on work using long-term datasets with a range of SSW events we will analyze the different qualities of RO and (re)analysis datasets more closely, including for different RO processing and (re)analysis variants.

## 3.2 Spatial and temporal variations of RO anomalies

Figure 5 shows distributions of MSTA, USDA and USTA anomalies over $50\,°-90\,°$N on four exemplary days of Jan 16, Jan 23, Jan 30, and Feb 13, 2009, depicting the space-time dynamics of the SSW event during different phases of its evolution. Looking at MSTA results, temperature anomalies are generally negative in most of the regions on Jan 16 with values up to $-30$ K. Positive anomalies emerge over the northern part of Atlantic Ocean ($0\,°-60\,°$W, $50\,°-55\,°$N). From that day on, positive anomalies move towards to higher latitudinal regions (can be seen from map results of other days not shown here, and from tracking of TEAs AM values discussed in Sect. 3.3 below). The magnitudes of the anomalies increase and the area of warming enlarges during the week after. This indicates an increase of the strength of the warming.

On Jan 23, positive temperature anomalies dominate the whole polar-cap region across the Atlantic sector, from over North America to over Europe. The warmest region is found centered on Greenland with anomalies exceeding 50 K. Results in this section (and in Sect. 3.3 below) indicate that Jan 23, 2009, is the warmest day of this SSW event. With the further progression of time, positive anomalies decrease, indicating a decrease of the strength of the warming. On Jan 30, smaller temperature anomalies up to 20 K are found. On Feb 13, which is two weeks after the warmest day, negative anomalies up to $-20$ K are found. Results of the LMBA (not shown) confirm that variations of bending angle anomalies are generally consistent with temperature anomalies, confirming the capability of RO bending angle to serve as a valuable support variable for monitoring SSWs, since this RO variable is observed accurately to better than 1 % up to about 60 km altitude (cf. Sect. 3.1).

USDA results, which are well suited to capture the downward propagated positive anomalies, show largest anomalies at the end of January. The warmest region is found from over Eastern Greenland to oceanic regions north of Russia. On Feb 13, large positive anomalies still occupy most of the polar region indicating a long-lasting warming effect caused by the SSW. The USTA results show negative (cooling) anomalies on the initial two days illustrated. However, on Jan 30, cooling anomalies are found to occupy most of the polar region. On Feb 13, the magnitude of the cooling anomalies increase to more than $-50$ K and the area of strong cooling is enlarged. This indicates a strong upper stratospheric cooling, with maximum cooling centered over the Oceanic part north of Russia.

## 3.3 SSW detection and monitoring results

Figure 6 shows the temporal evolution of the MSTA-TEA, LMBA-TEA, LSTA-TEA, USDA-TEA, and USTA-TEA results that instructively exhibit the threshold exceedance area changes during the SSW event. The geographic tracking of maximum (positive and negative) anomaly (AM) values is also shown. MSTA-TEA and LMBA-TEA results (first two rows)





are generally of similar characteristics, with positive anomalies emerging from Jan 17/18 which then quickly increase to maximum values on Jan 22/23. MSTA-TEAs are found to be largest on Jan 22, amounting for threshold exceedance areas over 30, 40 and 50 K to 18, 9, and 4 Mio. km$^2$, respectively. LMBA-TEA values are found to be largest on Jan 23, with areas exceeding the 30 %, 40 %, and 50 % thresholds amounting to 18, 12, and 5 Mio. km$^2$.

After the maximum value day, both MSTA-TEA and LMBA-TEA quickly decrease to zero. Such quick increase and decrease of the two metrics further reflect the sudden and rapid warming character of the SSW. Before the sudden warming, both TEAs show negative (cooling) anomalies as a pre-cursor signal. LMBA shows larger cooling anomalies with the TEA exceeding –30 % amounting to about 20 Mio. km$^2$. The negative anomalies show a tendency of increasing and reaching maximum on Jan 11 and then gradually decrease in approaching the beginning of the sudden warming. After the sudden

warming, there is a phase of silence where no strong positive/negative anomalies (exceeding $\pm 30$ K / %) are found. At the end of February, negative anomalies of both metrics emerge again. The right panel shows the tracking of AM values, indicating the movement of warming and cooling centers. It can be seen that the warming was centered over the east of Greenland, covering Greenland entirely and extending from Western Norway to Eastern Canada. During the most warmed days, the center locations of MSTA-TEA and LMBA-TEA AM values are rather close.

LSTA-TEA and USDA-TEA results are generally consistent in their evolution pattern as well, with most warming days found near the end of January and early February. Compared to the sharp increase and decrease of positive anomalies of MSTA-TEA and LMBA-TEA, the increase and decrease of LSTA-TEA and USDA-TEA are smoother, with maximum warming days somewhat delayed. The numbers of days showing positive (warming) anomalies are more than for MSTA-TEA and LMBA-TEA, indicating a longer-lasting warming at the lower stratospheric altitude levels. The locations of AM

values of the warming anomalies are centered over Northern Russia. Negative (cooling) anomalies are found from early to middle January and are strongest over the oceanic part northeast of Russia. The USTA-TEA results, finally, show strong cooling anomalies from early February throughout the month until end of February (end of this demonstration study analysis period). From middle to end of February, the TEAs that exceed a cooling of –30, –40, and –50 K, respectively, amount to more than 15, 8, and 3 Mio. km$^2$. The cooling centers are found over the oceanic part north of Russia. These results are

consistent to the strong upper stratospheric cooling in the SSW trailing phase found by a range of previous studies (e.g., Manney et al., 2008; Dhaka et al., 2015; Hitchcock and Shepherd, 2013).

Figure 7 depicts the overall results for our SSW metrics that we suggest to practically use for the detection and monitoring of SSW events. Geographic tracks of the metric-relevant temperature anomalies are shown as well. The first day on which the primary-phase metric SSW-PP-TEA exceeding 3 Mio. km$^2$ is Jan 20. From this day on, SSW-PP-TEA increases quickly up

to maximum on Jan 23 and then quickly decreases to be smaller than 3 Mio. km$^2$ on Jan 27. The secondary-phase metric SSW-SP-TEA first exceeds 3 Mio. km$^2$ on Jan 24 and then gradually increases to maximum on Jan 31 and gradually decreases to be smaller than 3 Mio. km$^2$ on Feb 8. SSW-PP-TEA and SSW-SP-TEA comprise our defined main-phase, i.e., where either or both of these two metrics exceed 3 Mio. km$^2$. The number of days of this main-phase, our defined main-phase duration SSW-MPD, is found 19 days for this Jan-Feb 2009 demonstration event. The mean TEA over the main-phase



duration, our defined main-phase area SSW-MPA, is $7.72\,\text{Mio.}\,\text{km}^2$ for this event. Multiplying duration and area yields the SSW's main-phase strength SSW-MPS, amounting to $146.6\,\text{Mio.}\,\text{km}^2$ days for this event. This clearly highlights it as a very strong SSW event that extended over an area of near $2000\,\text{km}$ effective radius around center location for more about two weeks; a major part of the polar north of $50\,°\text{N}$.

Summarizing relevant definitions, the first day of the main-phase is defined as the start day of the detected event and the end of the main-phase is defined as its final day. The center day is defined as the day with maximum TEA value of the primary metric, i.e., the Jan 23 of this demonstration event. The trailing metric SSW-TP-TEA (blue in Fig. 7), is an auxiliary metric to capture the long-lasting upper stratospheric cooling in the wake of the event. For this Jan-Feb 2009 event, the SSW-TP-TEA exceeds $3\,\text{Mio.}\,\text{km}^2$ from Feb 5, then gradually increases to a maximum of near $10\,\text{Mio.}\,\text{km}^2$ around middle February,

and then gradually decreases to $8\,\text{Mio.}\,\text{km}^2$ at the end of the study period (end of February).

As introduced in Sect. 2.3, a simplified fallback of the approach is to use temperature as the only variable for the metric estimation. Hence we illustrate in Fig. 7 also the results, where the primary- and secondary-phase metrics are computed from temperature only (the trailing-phase metric is temperature-only anyway). These two simplified metrics are generally seen consistent with the preferred dual variable-based metrics, but it is visible that they appear somewhat more "volatile" and less

robust in the sense that they exhibit more short-scale time variation. Follow-on work for a longer-term data record with a range of SSW events will analyze these characteristics in more detail.

The right panel shows that the main warming tracked by SSW-PPT-TEA (red) emerges from Norway and extends to Greenland and moves toward to higher latitudinal regions. The lower stratosphere warming (yellow/orange), tracked by SSW-SPT-TEA, is found emerging at the high latitudinal regions of Greenland and moving towards the northern part of

Russia. The upper stratospheric cooling (blue) tracked by SSW-SPT-TEA is found mainly at the high latitudinal oceanic region north of Russia.

These detection and monitoring results have been cross-tested using RO-collocated profiles from ECMWF analysis and also the regularly sampled ECWMF analysis fields as alternative data sources for these datasets. The results from both datasets (not separately shown) are found generally consistent for this demonstration event with the detection results using RO data.

This indicates that, on an individual SSW event basis, RO observational data are of comparable utility as ECMWF (re)analysis data to monitor the event and the influence of sampling uncertainty is small. This verifies that the new approach is readily applied to both observational and (re)analysis data (and also model output data). As discussed in the introduction (Sect. 1) and along with the analysis data description (Sect. 2.2), follow-on work on long-term records next needs to show how the possible advantages in long-term stability and accuracy of the RO data play out or not in SSW detection and

monitoring in comparison to reanalysis data.





# 4 Conclusion

In this study, we introduced a new approach to detect and monitor SSW events based on RO temperature, density, and bending angle anomaly profiles over $50\,°N - 90\,°N$ and demonstrated it for the well-known January-February 2009 event. The approach tracks the evolution by daily updates and is shown equally applicable to gridded (re)analysis data and, given

the same type of gridded field structure, also to model output data.

Based on constructed anomaly profiles for the three variables temperature, density, and bending angle, we employed the concept of Threshold Exceedance Area (TEA), which is the geographic area wherein absolute or relative anomaly values exceed predefined threshold values, as the basis for formulating SSW metrics. Computing TEAs based on anomalies in selected stratospheric altitude layers and using adequate threshold values (mainly $40\,K/40\,\%$), we formulated three SSW

detection and monitoring metrics. As a simplified fallback, the metrics can be computed alternatively from profiles or fields of temperature only.

The primary-phase metric is to examine the initial main-phase of warming caused by SSW events. The secondary-phase metric is to examine the further main-phase of downward propagated warming effects during the SSW. The trailing-phase metric is an auxiliary metric to co-examine the upper stratospheric cooling in the wake of an SSW. Based on the two main-

phase metrics, we introduced three key indicators for SSW detection and monitoring. The first is the main-phase duration, recording the number of days of SSW warming that exceed a defined minimum TEA (initially set to $3\,Mio.\,km^2$, corresponding to an area of about $1000\,km$ effective radius around center location). The second is the average daily main-phase TEA during main-phase duration, which is to quantify the average spatial extent of the event. The third is the area-duration product of the first two, termed main-phase strength, which expresses the overall strength and severity of the event.

For complementary space-dynamics information, the approach also enables, for the selected anomaly variables, daily tracking of the maximum anomaly values and of the related geographic center location of the event. In combination with the daily TEA estimates this quantifies also the approximate effective radius of the SSW-induced anomalies around the center location.

Applying the new approach for demonstration to the Jan-Feb 2009 SSW event, the detection and monitoring results find,

where it is comparable, similar characteristics as previous studies using other approaches and datasets. We found that the SSW warming emerged from about Jan 17 and reached maximum on Jan 23 and then fading by Jan 27. In terms of our three indicators, the duration of the main-phase of this SSW was 19 days, with an average main-phase area of $7.72\,Mio.\,km^2$, yielding main-phase strength of $146.6\,Mio.\,km^2$ days. This clearly highlights it as a very strong SSW event, for which pronounced anomalies $(>40\,K/>40\,\%)$ extended over an area of near $2000\,km$ effective radius around center location for

about two weeks; a major part of the polar cap north of $60\,°N$. The geographic tracking of the SSW showed that it was centered over East Greenland, covering Greenland entirely and extending from Western Norway to Eastern Canada. Cross-check application of the approach using ECMWF analysis data showed results generally consistent with these results from



RO data. This verifies the approach to be readily applied to both irregular profile-based observational, and to regular grid-based (re)analysis and model data.

Based on the encouraging demonstration in this study, follow-on work will apply the method to long-term RO and reanalysis datasets (RO overlapping $2006 - 2019$ with reanalyses over $1979 - 2019$) and assess its utility for long-term SSW

monitoring. In this way, the most suitable settings to use for the duration, area, and overall strengths indicators for robust SSW detection, monitoring, and classification can be determined. In addition, we will be able to learn how the possible advantages in long-term stability and accuracy of the RO data play out or not in SSW monitoring in comparison to reanalysis data, including for different variants of RO processing and reanalysis. Overall, we expect the approach to be valuable for monitoring how SSW characteristics unfold event by event but also, and in particular, how they possibly vary under transient

climate change and how they tele-connect to lower latitude regions.

*Author contributions.* Ying Li implemented the new method, performed the analysis, produced the figures, and wrote the initial draft of the manuscript. Gottfried Kirchengast served as primary coauthor, providing advice and guidance on all aspects of the design, analysis, figure production, and significantly contributed to writing of the manuscript. Marc Schwärz

supported the setup and advancements of the OPSv5.6 analysis system and advised on data and algorithms. Florian Ladstädter supported RO climatology provision and use and advised on data and the algorithm as well as on the results interpretation. Yunbin Yuan advised on analysis and algorithm comparison. All authors commented on the final submitted manuscript.

*Competing interests.* The authors declare that they have no conflict of interest.

*Acknowledgements.* We acknowledge ECMWF (Reading, UK) for providing access to their analysis and forecast data. The research at APM (Wuhan, China) funded by the National key Research Program of China "Collaborative Precision Positioning Project" (No. 2016YFB0501900), and the Chinese Natural Sciences Foundation (grant no. 41874040, 41604033).

At the WEGC (Graz, Austria) the work was supported by the Aeronautics and Space Agency of the Austrian Research Promotion Agency (FFG-ALR) under the Austrian Space Applications Programme (ASAP) project ATROMSAF1 (proj.no. 859771) funded by the Ministry for Transport, Innovation, and Technology (BMVIT).

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



**Table 1.** Basic parameters and methodology of the new SSW monitoring approach (all parameters (4)–(18) updated daily).

| Parameter | Equation/Definition | Explanation/Description |
|---|---|---|
| (1) Temperature anomaly profile $T_{\text{Anomaly}}$ | $T_{\text{Anomaly}} = T_{\text{RO}} - T_{\text{ROC1i}}$ | $T_{\text{RO}}$: RO temperature profile ;$T_{\text{ROCli}}$: collocated climatological profile |
| (2) Density anomaly profile $\rho_{\text{Anomaly}}$ | $\rho_{\text{Anomaly}} = (\rho_{\text{RO}} - \rho_{\text{ROC1i}}) / \rho_{\text{ROC1i}} \times 100\%$ | $\rho_{\text{RO}}$: RO density profile;$\rho_{\text{ROCli}}$: collocated climatological profile |
| (3) Bending angle anomaly profile $\alpha_{\text{Anomaly}}$ | $\alpha_{\text{Anomaly}} = (\alpha_{\text{RO}} - \alpha_{\text{ROC1i}}) / \alpha_{\text{ROC1i}} \times 100\%$ | $\alpha_{\text{RO}}$: RO bending angle profile; $\alpha_{\text{ROCli}}$: collocated climatological profile |
| (4) Middle Stratosphere Temperature Anomaly Threshold Exceedance Area: **MSTA-TEA** | Altitude range: 30–35 km<br>Thresholds selected:<br>+50 K, +40 K, +30 K; –30 K, –40 K, –50 K | Extract from individual anomaly profiles in selected stratosphere and stratopause region altitude layers (e.g., 30–35 km for MSTA-TEA) to estimate a vertical mean anomaly value for all RO events. The vertical mean anomalies are then averaged into a suitable space-time-binned grid over 50–90 °N (5 ° latitude × 20 ° longitude grid). The geographic areas wherein temperature, density and bending angle anomalies exceed predefined thresholds such as 40 K or 40 % are calculated and denoted as Threshold Exceedance Areas (TEAs). |
| (5) Lower Mesosphere Bending angle Anomaly Threshold Exceedance Area: **LMBA-TEA** | Altitude range:50–55 km<br>Thresholds selected:<br>+50 %, +40 %, +30 %; –30 %, –40 %, –50 % | |
| (6) Lower Stratosphere Temperature Anomaly Threshold Exceedance Area: **LSTA-TEA** | Altitude range: 20–25 km<br>Thresholds selected:<br>30 K, 25 K, 20 K;-20 K, -25 K, -30 K | |
| (7) Upper Stratosphere Density Anomaly Threshold Exceedance Area: **USDA-TEA** | Altitude range: 40–45 km<br>Thresholds selected:<br>50 %,40 %, 30 %; -30 %, -40 %, -50 % | |
| (8) Upper Stratosphere Temperature Anomaly Threshold Exceedance Area: **USTA-TEA** | Altitude range: 40–45 km<br>Thresholds selected:<br>+50 K, +40 K, +30 K; –30 K, –40 K, –50 K | |
| (9) Primary-phase metric: **SSW-PP-TEA** | SSW-PP-TEA [km²] = Avg(MSTA-TEA>40K, LMBA-TEA>40%) | Expresses the main and primary stratospheric warming anomaly strength |
| (10) Secondary-phase metric: **SSW-SP-TEA** | SSW-SP-TEA [km²] = Avg(LSTA-TEA>25K, USDA-TEA>40%) | Expresses the secondary downward propagated warming anomaly strength |
| (11) Trailing-phase metric: **SSW-TP-TEA** | SSW-TP-TEA [km²] = Abs(USTA-TEA<–40K) | Expresses the trailing upper stratosphere cooling anomaly strength |
| (12) Primary-phase T-only metric: **SSW-PPT-TEA** | SSW-PPT-TEA [km²] = (MSTA-TEA>40K) | Complementary primary metric using only temperature information |
| (13) Secondary-phase T-only metric: **SSW-SPT-TEA** | SSW-SPT-TEA [km²] = (LSTA-TEA>25K) | Complementary secondary metric using only temperature information |
| (14) Main-phase duration: **SSW-MPD** | SSW-MPD [days]<br>(definition see right column) | Number of days with SSW-PP-TEA or SSW-SP-TEA > TEA$_{\text{Min}}$ (3 Mio. km²) |
| (15) Main-phase area: **SSW-MPA** | SSW-MPA [Mio. km²]<br>(definition see right column) | Mean daily Max(SSW-PP-TEA, SSW-SP-TEA) during all SSW-MPD days |
| (16) Main-phase strength: **SSW-MPS** | SSW-MPS [Mio. km² days] = (SSW-MPA x SSW-MPD) | Overall strength, the larger this area-duration product, the stronger the event |
| (17) Anomaly Maximum (AM) values | $\Delta T_{\text{Max}}$ [K], $\Delta\alpha_{\text{Max}}$ [%], $\Delta\rho_{\text{Max}}$ [%] | Maximum (positive/negative) anomaly values of all grid cells over 50 °N-90 °N |
| (18) Geographic location (Lat, Lon) of AM values | $\varphi^{\text{AM}}$ [ °N], $\lambda^{\text{AM}}$ [ °E] | Generate a contour that is 2 K / 2 % smaller/larger than the positive/negative AM value; the center of the contour is then used as location of the AM value |

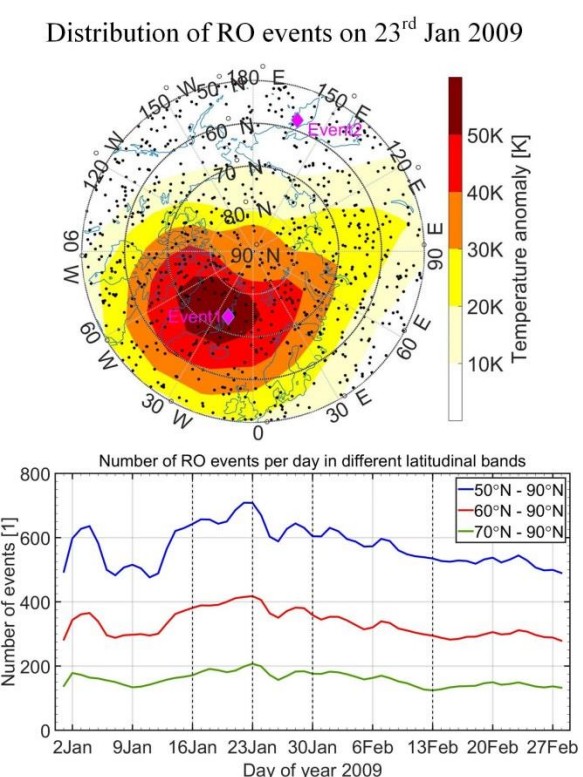

**Figure 1.** Illustrative distribution of RO event locations on 23 Jan 2009 (black dots), overplotted on the middle-stratosphere temperature anomaly of the day (upper panel), and number of RO events per day in the latitudinal bands of 50 – 90 °N (blue), 60 – 90 °N (red), and 70 – 90 °N (red), during January and February of 2009 (lower panel). In the upper panel, "Event1" represents an RO event with a large temperature anomaly, and "Event2" one with small anomaly (diamond symbols), as used in the subsequent Figs. 2 and 3.

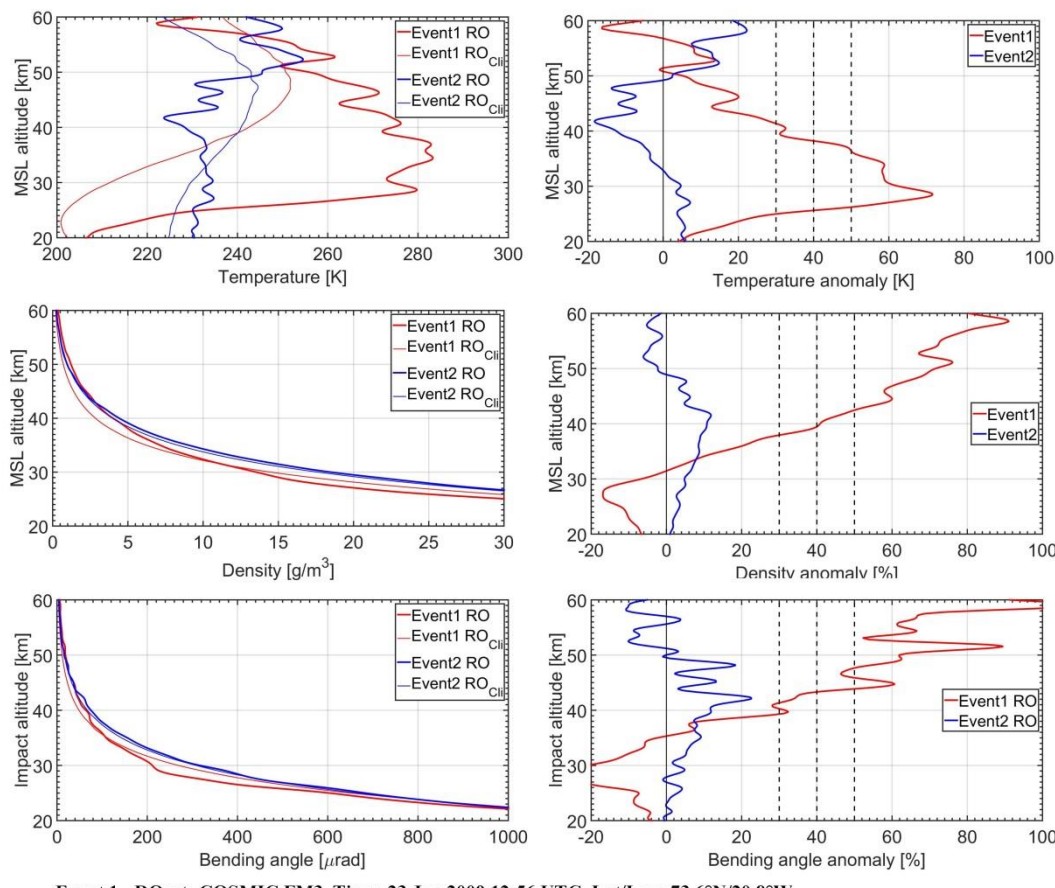

Event 1 - ROsat: COSMIC FM3, Time: 23 Jan 2009 12:56 UTC, Lat/Lon: 73.6°N/20.8°W
Event 2 - ROsat: COSMIC FM1, Time: 23 Jan 2009 06:02 UTC, Lat/Lon: 57.6°N/161.3°E

**Figure 2.** Event1 and Event2 temperature (top), density (middle), and bending angle (bottom) profiles from RO and their collocated climatological profiles $RO_{Cli}$ (left column), together with the corresponding anomaly profiles (right column), the latter computed according to Table 1, (1) – (3).



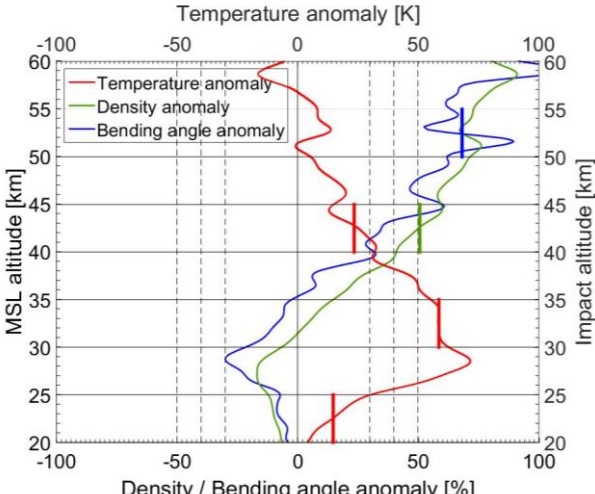

**Figure 3.** Temperature (blue), density (green) and bending angle (red) anomaly profiles of Event1 (same as in Figs. 1 and 2), with the horizontal gray lines delineating the altitude layers chosen for calculating the five basic TEAs (Table 1, (4) – (8)) and the colored vertical thick lines indicating the vertical mean anomaly values in corresponding altitude layers.

**Figure 4.** Temporal evolution of polar-cap (60° – 90°N) mean temperature (top), density (middle), and bending angle (bottom) anomaly profiles from RO (left column), complemented by the corresponding anomalies from the ECMWF analysis data (right column). The vertical dashed lines indicate four days selected for showing anomaly distributions in Fig. 5 and the horizontal lines in the panels delineate those altitude layers chosen for the respective variables to help compute the TEA metrics as presented in Fig. 3.



**Figure 5.** Middle Stratosphere Temperature Anomaly (MSTA, left column), Upper Stratosphere Density Anomaly (USDA, middle column) and Upper Stratosphere Temperature Anomaly (USTA, right column) on the four exemplary days of Jan 16, Jan 23, Jan 30, and Feb 13, 2009, illustrating the space-time dynamics of the SSW event in these three anomaly quantities.



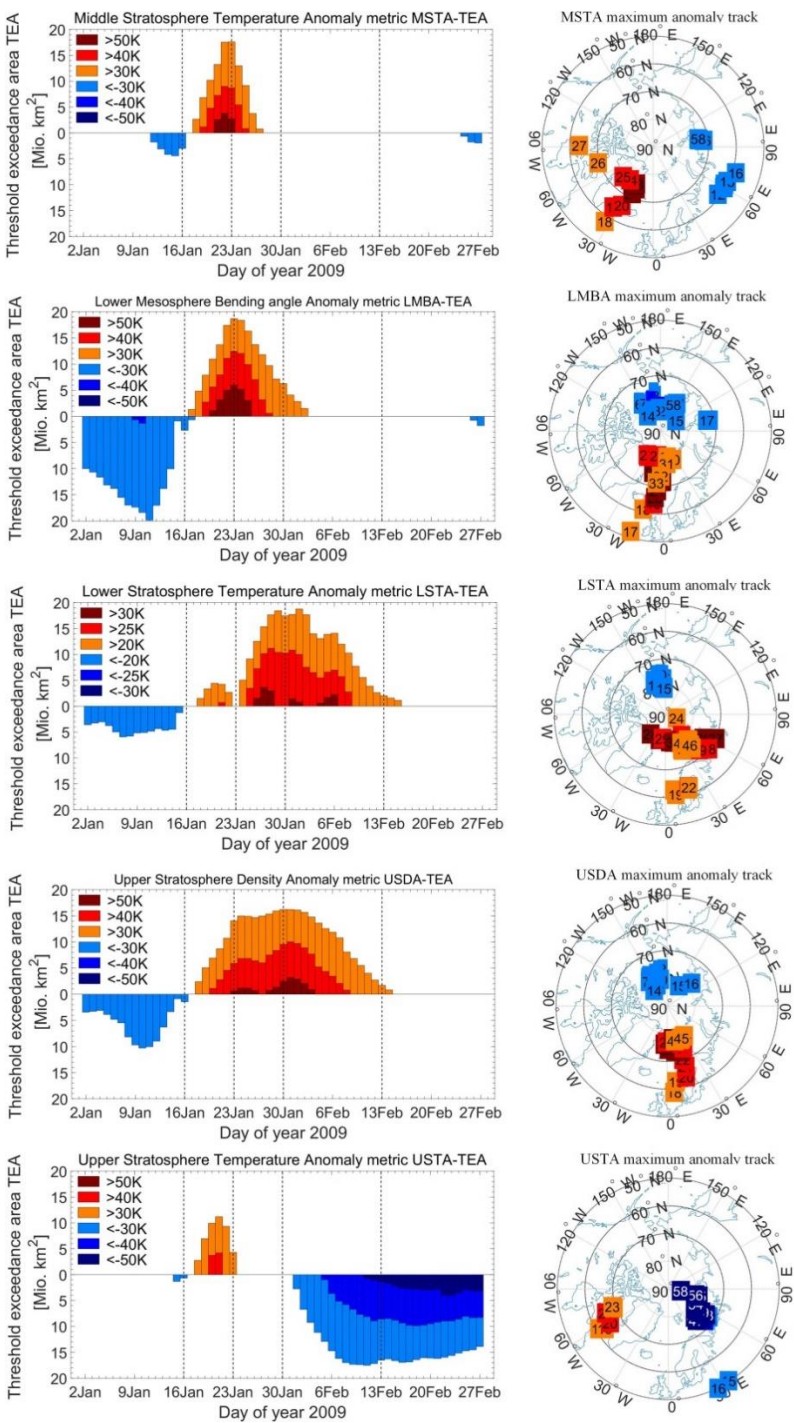

**Figure 6.** Time evolution of the daily MSTA, LMBA, LSTA, USDA, and USTA (from top to bottom) Threshold Exceedance Areas (TEAs) during the SSW event, using thresholds according to Table 1, (4)–(8) (left column). For complementary space-dynamics information, geographic tracks and magnitude classes (color scheme of left panels, numbering by day-of-year) of maximum positive/negative anomaly values are shown (right column).



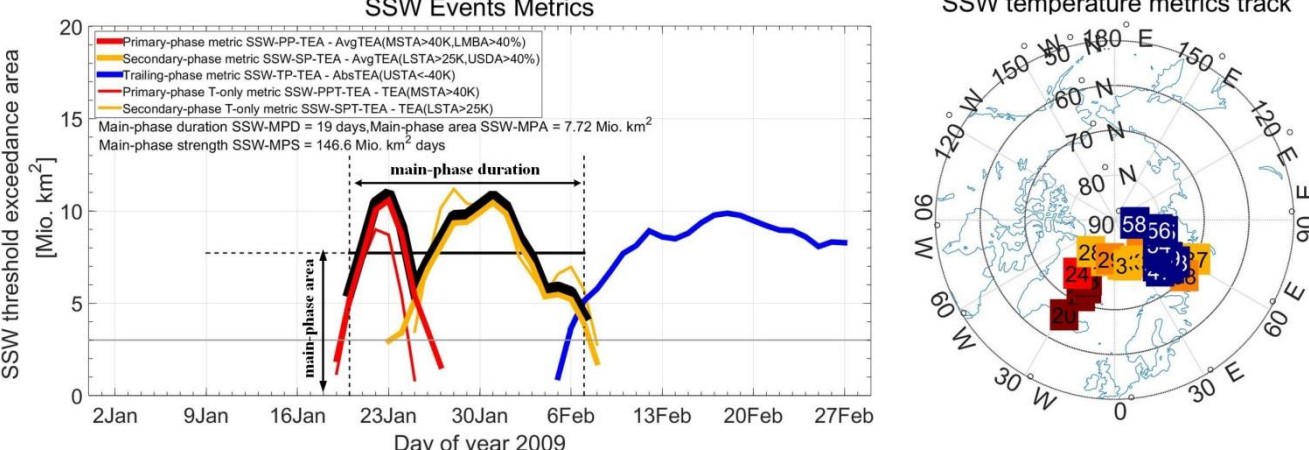

**Figure 7.** Time evolution of the daily primary-phase (heavy red), secondary-phase (heavy yellow), trailing-phase (heavy blue), primary-phase temperature-only (light red), and secondary-phase temperature-only (light yellow) metrics, respectively (left panel), shown for daily TEAs exceeding $1\,\text{Mio.}\,\text{km}^2$. The main-phase metrics envelope for computing the main-phase area and duration (heavy black) and the related area, duration, and strength indicator results are depicted as well (the numeric results in legend) and the TEA$_{Min}$ threshold of $3\,\text{Mio.}\,\text{km}^2$ is indicated as gray horizontal line. For complementary space-dynamics information, the geographic tracks and magnitude classes of the three metric-relevant temperature anomalies (MSTA red, LSTA yellow, USTA blue) are also shown (right; style as in right panels of Fig. 6).