# Peer review of "Monitoring Sudden Stratospheric Warmings using radio occultation: a new approach demonstrated based on the 2009 event"

_Atmospheric Measurement Techniques, 2020_

## Referee Comment (RC1) · Anonymous Referee #3 · 8 Aug 2020

This paper proposes a new application of GNSS-RO for detecting stratospheric sudden warming (SSW) events. I appreciate the devoted efforts of the authors for developing an interesting analysis technique with the COSMIC GNSS-RO data and applying it to the major SSW event that occurred in 2009. The results are impressive in visualizing the horizontal distribution as well as the time evolution of the 2009 SSW event. However, I have several concerns about the data analysis procedure and the usefulness of the proposed techniques.

Major comments:

1. This method is successfully applied to the SSW event of January 2009, which

is a well-studied case. I am afraid that Section 3 is too descriptive. For comparing the analyzed results with earlier studies, any new scientific findings on SSW behavior should be reported. The technique should be evaluated, showing any new features of the SSW that can be uniquely resolved by GNSS-RO data.

2. Before proposing this method to monitor major/minor SSW events from long-term records, more cases should be tested to confirm that it is fully robust.

3. SSW is defined by the temperature anomaly in this study, but it is also characterized by zonal wind reversal. As GNSS-RO can provide only the former information, it may not clarify the entire behavior of SSW; therefore, this method may not be considered as primary. (See comment 1.)

4. The analysis procedure is a bit complicated. It employs three anomaly parameters: temperature, density, and bending angle, in four altitude ranges. Out of a total of 12 values, only five parameters listed in Table 1 are used to monitor the SSW characteristics. It is not clear whether the selection of these five parameters will generally be adopted for any SSW event, or this set is used specifically for the 2009 SSW event.

5. Assuming that the COSMIC GNSS-RO data is assimilated into ECMWF, the 2009 SSW naturally appears similar in both GNSS-RO and ECMWF. Therefore, agreement of the SSW characteristics, as shown in Fig. 4, does not necessarily confirm the validity of the proposed GNSS-RO method.

6. The accuracy of the GNSS-RO data in the upper stratosphere and mesosphere (above about 40–50 km) should be tested carefully, because the error in the bending angle due to ionospheric effects could dominate, depending on the ionospheric conditions. Moreover, it is noteworthy that the bending angle profile at high altitudes is heavily optimized by referring to a model atmosphere profile, reducing the deviations from a climatological profile.

7. As the analyzed values are a weighted mean over three days, the time resolution

is longer than one day. Therefore, the time evolution of the SSW event, such as its duration and onset date, cannot be precisely determined at a daily resolution.

8. I would encourage the authors to extend the latitude range below 60°N, as the effects of SSW on the middle latitudes and equatorial regions have been the subject of recent research. I would also suggest the use of ionospheric electron density data with GNSS-RO to identify SSW effects on the upper atmosphere.

Specific comments:

9. P2, L27–30: Show some references on the limitations of other satellite missions.

10. P3, L1–3: Similarly, explain the limitations of the reanalysis data, referring to the relevant papers.

11. P5, L19–25: GNSS-RO data are neither distributed evenly nor regularly, but randomly with a relatively high horizontal density.

12. P6, L23–25: Is the GNSS-RO data assimilated into the ECMWF? If so, the agreement of the climatology is reasonable. (See comment 5.)

13. P6, L32: Remove one of "the".

14. P7, L9–11: Temporal resolution is lower than one day, which affects the description of the time evolution of SSW, such as its duration and onset date. (See comment 7.)

15. P7, L14–17: For the four altitude regions, the exact height ranges should be provided here, even though they are shown in Table 1.

16. P7, L30–34: Is selection of the thresholds intended to be applicable to any SSW events, or specific to the 2009 case? (See comment 4.)

17. P8, L2: Isn't 50% of the density deviation reasonable? It seems too large. (See comment 6.)

18. P8, L13: The word "then" can be read as "than", right?

19. P8, L19–20: "number of days". (See comment 14.)

20. P8, L22–23: How is the technique adjusted for long-term data? It sounds like this method is not fully robust, and a specific tuning is required for each SSW event. (See comment 4.)

21. P9, L26–28: "differences above 50 km". (See comment 6.)

22. Section 3, P11, L24–26: I encourage the authors to show any new scientific findings obtained with the GNSS-RO data. (See comments 1 and 3.)

---

## Referee Comment (RC2) · Anonymous Referee #1 · 14 Aug 2020

This is a well-done manuscript outlining the details of and application of a novel method for detecting and evaluating sudden stratospheric warmings (SSWs). The authors aptly couch their work in the context of the ongoing discussion within the SSW community about SSW definitions. They demonstrate that their method and definitions, at least for the 2009 SSW, agree with established metrics and provide additional objective information. While the context of their work is centered around the use of radio occultation data, the authors show that using a selected model's data results in complimentary analysis, showing that this work may readily be applied to long-term reanalyses.

I find this work to be properly placed in the literature and a novel contribution to the

community. I do have a few comments I would like the authors to address prior to publication.

Minor comments

1: My main concern about the manuscript is on how clear the authors are in letting the reader know that the particular threshold choices are determined based on this one anomalous event. I appreciate that they do make this clear in the conclusions section, but that clarity was missing in Section 2.3 where the threshold values are introduced. In particular, I think the paragraph beginning on page 7, line 27 could use an additional statement(s) on this topic.

Along these lines, I think some additional clarity in the statement on pg. 12, line 1 is warranted. Certainly, this SSW is known for being strong, but as-written, the authors seem to suggest that their method is sufficient to determine that this event is strong. Given that the work in this manuscript is based off a single SSW, it's not obvious how that can be determined independently of other SSWs.

I think the authors should critically consider other areas of the text that would benefit from further discussion about this topic.

2: The authors bring up the Butler et al. (2015) requirements for a standard definition of SSWs. Missing from the manuscript is the authors' discussion on how their definition fits these three proposed criteria. These are criteria the SSW community has agreed upon, so providing additional contextualization of their method in light of these should be done.

Specific comments

1: Do the authors report somewhere that bending angle and density are given in normalized units? This is apparent, but the reader would benefit from a definitive statement in the manuscript. As well, please state how the normalization is performed (normalized with respect to what?).

2: Abstract, line 20: recommend "has strong potential."

3: Abstract, lines 17 and 22: is it necessary to introduce these metrics – the 3 Mio. km^2 threshold and the MSTA-TEA40 metric – here? I'm not sure that the abstract benefits from either the specificity of the former or the raising of the as-yet undefined metric and abbreviation of the latter. I would recommend removal unless the authors have strong objections.

4: Pg. 8, lines 25-26: I'm not quite sure I follow what's being said here. Is it that the specific definitions the authors have proposed may change as more systematic study is performed?

---

## Author Comment (AC1) · 15 Oct 2020

**Response to Reviewer 3 Comments**

***General comments****: This paper proposes a new application of GNSS-RO for detecting stratospheric sudden warming (SSW) events. I appreciate the devoted efforts of the authors for developing an interesting analysis technique with the COSMIC GNSS-RO data and applying it to the major SSW event that occurred in 2009. The results are impressive in visualizing the horizontal distribution as well as the time evolution of the 2009 SSW event. However, I have several concerns about the data analysis procedure and the usefulness of the proposed techniques.*

**Overall response:** We thank Reviewer 3 for his/her comments. We have carefully addressed all the comments as stated below and also in the Revised Manuscript (RM).

***Major comments:***

***Point 1****: This method is successfully applied to the SSW event of January 2009, which is a well-studied case. I am afraid that Section 3 is too descriptive. For comparing the analyzed results with earlier studies, any new scientific findings on SSW behavior should be reported. The technique should be evaluated, showing any new features of the SSW that can be uniquely resolved by GNSS-RO data.*

**Response 1:** Thanks for the comments. As we have discussed in the introduction section of this paper, one key motivation of this initial paper is to pave the way towards an improved standard definition of SSW, and such a definition does not exist yet due to lack of sufficiently reliable observational data and also the diversity of application purposes. Therefore, we intend to propose a robust method to detect SSW and subsequently to propose a standard definition of SSWs.

For our initial study here, we used this typical 2009 SSW event. The new method shows its basic potential for reliable SSW detection. The results obtained using this new method are consistent with results shown by previous researches. We will apply this new method in future to longer-term data records, and have just very recently started this work, to refine our specific settings and also thresholds to finally make a broadly useful SSW detection method. Based on this we also intend to introduce a standard definition of SSWs.

The prime new features of this study are hence in our method design. It can satisfy the requirements that were outlined by Butler et al. (2015), reading as follows (cited from line 33 to 34 in page 3 and from lines 1 to 2 in page 4 in the RM):

"*The new definition should be proposed primarily for the purpose of describing polar winter variability. Secondly, it should be easily calculated and applicable to reanalysis and model outputs, both in post-processing and in real time. Finally, the new definition should not be highly sensitive to details, such as an exact latitude, background climatology, threshold wind speed, spatial extent, or pressure level.*"

Our proposed method can describe the variability over all polar region before, during

and after the occurrence of SSWs and also over different stratospheric layers. The new features of our new method include that:

(1) It can be applied to RO and similar profile data, and also gridded reanalysis data such as from ECMWF.

(2) It uses anomalies over selected height layers and also introduces the concept of TEAs, robustly quantifying warming or cooling areas in polar region. This design makes our method properly insensitive to any perturbing fine details.

(3) RO bending angles, and density profiles, are found useful auxiliary variables for SSW detection, but it similarly works for temperature-only data as well.

Based on our new method, Section 3 also shows all new findings obtained using our new method, including our metrics, TEAs and also RO bending angles anomalies during SSW, which have not been analyzed by any other previous researchers.

To make our new features clearer for readers to comprehend, as also suggested by Reviewer 1, we have added some discussions at the end of Section 3 (from lines 8 to 15 in page 13 in the RM), as follows:

*"To summarize, the metrics proposed in this study for monitoring the SSW events can well satisfy the conditions that Butler et al. (2015) suggest for proposing a standard definition (cf. Section 1). Firstly, our approach well captures the sudden warming of the main phase and also its downward propagation into the lower stratosphere as well as the cooling occurring after the warming phase in the upper stratosphere. Secondly, the approach can be used for both RO and other suitable profile data and likewise for reanalysis data, and can be applied for both post-processing and in real time. Finally, the new approach is using anomalies over several height layers, and TEAs over larger area, and hence the detection and monitoring results are not sensitive to details such as exact latitude or pressure level. Potential further refinements of the thresholds for our metrics will be determined from recently started work on multiple SSW events, using longer-term data over the recent decades."*

***Point 2:*** *Before proposing this method to monitor major/minor SSW events from long-term records, more cases should be tested to confirm that it is fully robust.*

**Response 2:** Thanks for this comment. Yes, we will make sure this method is fully robust by applying this method to nearly 1.5 decades (14 years) of RO data and also several decades of ERA-5 data, as we stated in the original manuscript (and see now lines 16 to 18 in page 14 in the RM, where we updated that the winter half year 2019/20 is now as well already available):

*"Based on the encouraging demonstration in this study, follow-on work will apply the method to long-term RO and reanalysis datasets (RO overlapping 2006–2020 with reanalyses over 1979–2020) and assess its utility for long-term SSW monitoring."*

***Point 3:*** *SSW is defined by the temperature anomaly in this study, but it is also characterized by zonal wind reversal. As GNSS-RO can provide only the former*

*information, it may not clarify the entire behavior of SSW; therefore, this method may not be considered as primary. (See comment 1.)*

**Response 3:** Thanks for the comment. As what has been discussed by Butler et al. (2015, 2018) and also in this paper, there is currently no standard definition of SSWs. There are currently nine often used definitions, in terms of zonal-mean winds at 10 hPa and 60° latitude, geopotential height anomalies, and also temperature anomalies, etc. Among these definitions, SSWs characterized by zonal wind reversal is one of the common used definitions. However, this does not mean that SSW definitions must be characterized by wind reversals.

According to literature, the SSW phenomena was first noted by its sudden and quick temperature increase measured by radiosonde since the 1950s. Wind reversal is one of its important characteristics occurring during this sudden warming. It is a useful metric for SSW. However, we also noted that selection of certain altitude and also latitude for measuring of wind reversals can results in different detection results. The SSW climatology obtained from previous studies, by using wind reversals at 65°N and 60°N, could result in very different results in some cases, with the center day varying more than 2 weeks. From this, we can see that if characterizing SSWs by wind reversals, results may suffer from detailed selection of latitudes.

Our method, which is relying on anomalies over several selected altitude layers, and also large threshold exceedance areas (TEAs), is not sensitive to such detail selection. The method is also not only suitable for RO or similar profile data, and also allows for real-time applications, and can likewise be applied to reanalysis data. Since reanalysis data also provide wind information, users who prefer to use reanalysis data could also extract wind information together with our method for SSW detection.

For our treatment in the manuscript, we tried to indeed provide an informative introduction and overview of this (without excessive length); and please see also our response to comment 1.

**Point 4:** *The analysis procedure is a bit complicated. It employs three anomaly parameters: temperature, density, and bending angle, in four altitude ranges. Out of a total of 12 values, only five parameters listed in Table 1 are used to monitor the SSW characteristics. It is not clear whether the selection of these five parameters will generally be adopted for any SSW event, or this set is used specifically for the 2009 SSW event.*

**Response 4:** Thanks for the comment. For formulating our metrics, we select five TEAs based on five thresholds. These thresholds will be further refined after we apply our method to longer data records, a work we just recently started. The method will then can be adopted for any SSW event.

**Point 5:** *Assuming that the COSMIC GNSS-RO data is assimilated into ECMWF, the 2009 SSW naturally appears similar in both GNSS-RO and ECMWF. Therefore, agreement of the SSW characteristics, as shown in Fig. 4, does not necessarily*

*confirm the validity of the proposed GNSS-RO method.*

**Response 5:** Thanks for the comments. The proposed method does in fact not just rely on RO data. As noted in a response above already, it is a method that can use both RO data and/or analysis data from model, such as ECMWF data. If other observation data can offer global dense distributed and high quality atmospheric profiling data over the stratosphere, the data can also be applied to this method.

The reason we incorporate RO data, but not just the model data is because RO data can provide accurate profiling for climate-type monitoring, but also available in form of real-time observation of the atmosphere. Furthermore, RO data have several distinctive advantages, such as the high vertical resolution in stratosphere (~1 km), global coverage with very repeatable quality, and the availability of useful auxiliary variables such as RO bending angles. It is such advantages that make RO data quite suitable for SSW detection on their own.

The impacts of assimilation of RO data into ECMWF is that ECMWF analysis data have higher accuracy in regions, where RO data of high density and accuracy that other observation data. It does not affect the validity of our method to ECMWF data at all.

**Point 6:** *The accuracy of the GNSS-RO data in the upper stratosphere and mesosphere (above about 40–50 km) should be tested carefully, because the error in the bending angle due to ionospheric effects could dominate, depending on the ionospheric conditions. Moreover, it is noteworthy that the bending angle profile at high altitudes is heavily optimized by referring to a model atmosphere profile, reducing the deviations from a climatological profile.*

**Response 6:** In our original manuscript, though we used so-called optimized bending angle, we have internally tested that using the ionosphere-corrected atmospheric bending angles directly (i.e., non-optimized bending angles) are very similar to the results from using optimized bending angle. In order to avoid this confusion in our revised manuscript, we only used the atmospheric bending angles for the calculation of bending angle anomalies now, so that "optimization" cannot play any role. In this form, atmospheric bending angles are indeed independent observations not affected by background model atmospheric profiles. In order to avoid somewhat higher short-scale variability of these non-optimized bending angles, we use a 2 km vertical averaging for smoothing over the short-scale variations. We tested again, and found confirmed, that the results from using the atmospheric bending angles directly are similar to the ones using optimized bending angles. The RM includes these updates accordingly. The figures related to bending angle are all updated to use non-optimized bending angle. We have also used describe this in the text from lines 22 to 23 in page 6 as below:

"*In order to avoid the impacts of background model on bending angles, we used non-optimized bending angle in our study.*"

**Point 7:** *As the analyzed values are a weighted mean over three days, the time resolution is longer than one day. Therefore, the time evolution of the SSW event,*

*such as its duration and onset date, cannot be precisely determined at a daily resolution.*

**Response 7:** What we do (as we describe) is that we use a weighted mean over three days, just to allow more RO events for a reliable statistical calculation. However, in order to get our daily sampling to achieve a near-daily resolution, we give the data of the middle day a twice-high weight of 0.5, while the weights of the predecessor and successor days are only 0.25. In this way the daily sampling is indeed still meaningful, while we statistically improve the robustness of the average profiles from enabling a bigger ensemble of profiles for the weighted averaging, by taking 3-day windows.

We note also that we have made detail sensitivity tests to make sure that our selection of temporal resolution is robust. We have calculated anomalies by just using one day's data and we have also calculated anomalies by using same weight of all three days (each day 1/3), etc. We found that if just using one day's data, the magnitudes of the resulting anomalies are similar to our weighted mean over three days, but somewhat nosier. If using three days' mean with even weights for all days, the dynamic daily evaluation of SSWs would be somewhat more blurred. So at the end we chose, as a most suitable trade-off, the weighted averaging that favors the center day's weight as noted above.

In order to make this clearer also for readers, we have added the following statements from lines 16 to 19 in page 7 as below:

"*In this way the daily sampling is indeed still meaningful, while we statistically improve the robustness of the average profiles from enabling a bigger ensemble of profiles for the weighted averaging, by taking 3-day windows.*"

**Point 8:** *I would encourage the authors to extend the latitude range below 60_N, as the effects of SSW on the middle latitudes and equatorial regions have been the subject of recent research. I would also suggest the use of ionospheric electron density data with GNSS-RO to identify SSW effects on the upper atmosphere.*

**Response 8:** Thanks for the comment. However, in this initial paper we prefer not to study at the same time the effects of SSWs on the middle latitudes and equatorial regions, but rather focus on introducing a robust new SSW detection method. Therefore, we decided on this focus on polar region here, though in future we are indeed interested to analyze also "teleconnection effects" towards the lower latitudes. Similarly, using of ionospheric electron density data to identify SSW effects are also beyond the focus in this study.

**Specific comments:**

**Point 9:** P2, L27–30: Show some references on the limitations of other satellite missions.

**Response 9:** OK. We have added two new references, i.e., McInturff, et al. (1978) and

Manney et al. (2008), which have discussed the limitations of conventional satellites data. Please refer to line 29 amd 30 in page 2 in the RM.

***Point 10:*** *P3, L1–3: Similarly, explain the limitations of the reanalysis data, referring to the relevant papers.*
**Response 10:** OK. We have added the Butler et al. (2015) also as a reference here, which have discussed the limitations of the reanalysis data. Please refer to line 3 in page 3 in the RM.

***Point 11:*** *P5, L19–25: GNSS-RO data are neither distributed evenly nor regularly, but randomly with a relatively high horizontal density.*
**Response 11:** Thanks, yes. In order to make this description more accurate, we have updated the corresponding statements from lines 20 to 24 in page 5, now stated as follows:

*"The upper panel shows the distribution of RO observations in the study domain from 50° N to North Pole, within which strong warmings were found by previous studies of the SSW event (Labitzke and Kunze, 2009; Harada et al., 2010; Kodera et al., 2011; Taguchi et al., 2011). It can be seen that most of the polar region is covered by RO observations. Similarly, high observation density also applies to the other days of the study period."*

**Point 12:** P6, L23–25: Is the GNSS-RO data assimilated into the ECMWF? If so, the agreement of the climatology is reasonable. (See comment 5.)
**Response 12:** Thanks, yes. And as responded to Point 5, assimilation of GNSS-RO data does not affect the application of our method, which can be applied to both RO and ECMWF data.

**Point 13:** P6, L32: Remove one of "the".
**Response 13:** Thanks. We have deleted this "the" in the RM.

**Point 14:** P7, L9–11: Temporal resolution is lower than one day, which affects the description of the time evolution of SSW, such as its duration and onset date. (See comment 7.)
**Response 14:** Thanks. Following up to our response to Point 7 above, our strategy of three days' averaging by given the center day twice the weight does not appreciably affect the quantification of the time evolution of SSWs, based on our sensitivity tests. We have also applied our method to the regularly sampled ECMWF data (2.5*2.5 grid points and four time layers a day), without such three days' averaging for these data, but the conclusions are still the same as the other results discussed.

**Point 15:** P7, L14–17: For the four altitude regions, the exact height ranges should be provided here, even though they are shown in Table 1.
**Response 15:** Thanks. We have added the corresponding altitude ranges we used for these five TEAs (in lines 31 to 32 in page 7 in the RM).

**Point 16.** P7, L30–34: Is selection of the thresholds intended to be applicable to any SSW events, or specific to the 2009 case? (See comment 4.)

**Response 16:** These thresholds are currently applied to this 2009 case. We will refine these thresholds after we have applied our method to longer-term data records, which is a work just recently started. However, the method will be the same as now, but only with some refined threshold definitions.

We have discussed this in the original manuscript and now further updated the corresponding discussion (in lines 3 to 6 in page 9, in the RM and in lines 16 to 23 in page 14 in the conclusion part of the RM), as follows:

*"In a follow-on work using long-term RO and reanalysis datasets, these indicators will be used to detect SSW events, for example by requiring a minimum main-phase duration of 7 days or so to qualify as an SSW, and to record the strength of the events. However, the specific thresholds for our metrics and indicators for SSW detection, monitoring, and classification can only be determined after the new approach is applied to longer-term data containing multiple events."*

"*Based on the encouraging demonstration in this study, follow-on work will apply the method to long-term RO and reanalysis datasets (RO overlapping 2006–2020 with reanalyses over 1979–2020) and assess its utility for long-term SSW monitoring. In this way, the most suitable settings to use for the duration, area, and overall strengths indicators for robust SSW detection, monitoring, and classification can be determined.*
*In addition, we will be able to learn how the possible advantages in long-term stability and accuracy of the RO data play out or not in SSW monitoring in comparison to reanalysis data, including for different variants of RO processing and reanalysis. Overall, we expect the approach to be valuable for monitoring how SSW characteristics unfold event by event but also, and in particular, how they possibly vary under transient climate change and how they tele-connect to lower latitude regions.*"

**Point 17.** P8, L2: Isn't 50% of the density deviation reasonable? It seems too large. (See comment 6.)

**Response 17:** Thanks for the comments. We have double checked all data and computations and find the density anomaly magnitudes up to roughly these levels, which are also consistent with bending angle anomaly sizes. Since many existing studies show polar mean anomalies or anomalies smoothed over multi-day periods, the anomaly magnitudes are somewhat smaller with those stronger smoothings. Here we exploit regional mean anomalies which are based on relatively small bin areas (5°×20°) on a daily sampling basis, therefore, the magnitudes are somewhat larger. See also the comment appended at the end of this response document; a technical correction (using the correct long-term background climatology) also made the anomalies a bit smaller, though in general for strong SSW events like this one in 2009 they reach these magnitudes.

**Point 18.** P8, L13: The word "then" can be read as "than", right?

**Response 18:** Yes, thanks for pointing this out. We have corrected "then" to "than" in the RM (line 23 in page 8 in the RM).

**Point 19.** P8, L19–20: "number of days". (See comment 14.)

**Response 19:** The "number of days" here suggests the number of days that our main-phase records last. Regarding the concern about the daily temporal resolution, please see our response to Points 7 and 14 above.

**Point 20.** P8, L22–23: How is the technique adjusted for long-term data? It sounds like this method is not fully robust, and a specific tuning is required for each SSW event. (See comment 4.)

**Response 20:** We have not applied this method to longer-term data in this initial study, we just very recently started such work. From our researches on this 2009 event, and also some complementary cross-check analyses (on the polar-cap mean, on daily mean anomalies of other years, etc.) we found that our selection of thresholds is basically reasonable. Yet, as mentioned already above in our response to Point 2, in follow-on work we will apply our method to almost 1.5 decades (14 years) of RO data and several decades of ERA-5 data, to refine our specific thresholds and make our method fully robust for the detection of multiple SSW events of various strengths.

**Point 21.** P9, L26–28: "differences above 50 km". (See comment 6.)

**Response 21:** Please refer to our Response to Point 6.

**Point 22.** Section 3, P11, L24–26: I encourage the authors to show any new scientific findings obtained with the GNSS-RO data. (See comments 1 and 3.)

**Response 22:** Please see our extended response to Points 1 and 3. The method itself is all new so we prefer to keep the length and scope of this manuscript in its current form. We will be interested in such extended scopes in follow-on work. As to the new content of this manuscript, we note that Section 2 and also the results in Section 3 are all essentially new developments and findings of this paper.

We thank Reviewer 3 for his/her comments again. In addition to the comments and suggestions from the Reviewers, we inform that based on our careful rechecks of the full computation we found that a technical bug remained in our original calculation of the background climatology. It implied that we inadvertently had used the mean climatology of January and February of the year 2017 only, instead of the long-term mean climatology from the full range 2007 to 2017. We have corrected this in the updated RM.

We found that all our basic conclusions are still robust and valid; just the detailed quantifications changed, since the long-term climatology yields a somewhat different (and even more smooth) reference background. Figures 1 to 5 are very similar to the original ones. The main impact is that the TEAs from the original thresholds are now

smaller, while the TEAs temporal characteristics are strongly similar. Therefore, in the RM, we generally selected somewhat lower thresholds for calculating TEAs and for formulating our metrics, to keep their sizes and magnitudes rather similar. The Figures 6 to 7 reflect this and confirm that there was no qualitative change, just limited quantitative change that left the conclusions robust. Please find the RM for further details where we made small adjustments to the text accordingly.

---

## Author Comment (AC2) · 15 Oct 2020

**Response to Reviewer 1 Comments**

***Overall comments:*** *This is a well-done manuscript outlining the details of and application of a novel method for detecting and evaluating sudden stratospheric warmings (SSWs). The authors aptly couch their work in the context of the ongoing discussion within the SSW community about SSW definitions. They demonstrate that their method and definitions, at leastfl for the 2009 SSW, agree with established metrics and provide additional objective information. While the context of their work is centered around the use of radio occultation data, the authors show that using a selected model's data results in complimentary analysis, showing that this work may readily be applied to long-term reanalyses. I find this work to be properly placed in the literature and a novel contribution to the community. I do have a few comments I would like the authors to address prior to publication.*

**Overall Response:** We thank Reviewer 1 for his/her overall positive comments. We have carefully addressed all the comments as stated below and also in the Revised Manuscript (RM).

**Minor comments**

***Point 1:*** *My main concern about the manuscript is on how clear the authors are in letting the reader know that the particular threshold choices are determined based on this one anomalous event. I appreciate that they do make this clear in the conclusions section, but that clarity was missing in Section 2.3 where the threshold values are introduced. In particular, I think the paragraph beginning on page 7, line 27 could use an additional statement(s) on this topic.*

*Along these lines, I think some additional clarity in the statement on pg. 12, line 1 is warranted. Certainly, this SSW is known for being strong, but as-written, the authors seem to suggest that their method is sufficient to determine that this event is strong.*

*Given that the work in this manuscript is based off a single SSW, it's not obvious how that can be determined independently of other SSWs.*

*I think the authors should critically consider other areas of the text that would benefit from further discussion about this topic.*

**Response 1:** Thanks for these comments. Yes, we agree that it is an improvement to discuss some more details how we select our thresholds for calculating the five TEAs and for formulating the metrics. We have hence inserted more statements from lines 3 to 7 in page 8, and from lines 18 to 20 in page 8 in the RM, with text as follows:

"*As the thresholds for calculating these five TEAs, we use those defined in Table 1, (4)–(8); for example, the thresholds for MSTA are 30, 35, and 40 K as seen therein. The selection of these thresholds was mainly guided by results on the polar-mean and regional mean anomalies shown in Sections 3.1 and 3.2. We examined the temporal variations of the magnitudes of warming and cooling of the five TEAs by sensitivity checks and finally chose suitable thresholds as summarized in Table 1 for illustration*

*of this 2009 event.*"

"*The thresholds for formulating the metrics are selected based on the condition that the TEAs calculated for the chosen thresholds can suitably capture the main features of warming or cooling of the SSW event.*"

Yes, with the statement in the first line of page 12, we should not indicate that this event is a very strong or non-so-strong one, just based on this single event. Hence we modified this from lines 14 to 16 in page 12 in the RM, as follows:

"*In lines with previous studies on this particular event, and also on recent preliminary studies on several other events, the values of main-phase duration and also of the strength indicate that this is a very strong SSW event.*"

**Point 2:** *The authors bring up the Butler et al. (2015) requirements for a standard definition of SSWs. Missing from the manuscript is the authors' discussion on how their definition fits these three proposed criteria. These are criteria the SSW community has agreed upon, so providing additional contextualization of their method in light of these should be done.*

**Response 2:** Thanks for the comments. We have added discussions from lines 8 to 15 in page 13 in the RM to state how our method meets the requirements of standard definitions of SSWs. The updated statements are as follows:

"*To summarize, the metrics proposed in this study for monitoring the SSW events can well satisfy the conditions that Butler et al. (2015) suggest for proposing a standard definition (cf. Section 1). Firstly, our approach well captures the sudden warming of the main phase and also its downward propagation into the lower stratosphere as well as the cooling occurring after the warming phase in the upper stratosphere. Secondly, the approach can be used for both RO and other suitable profile data and likewise for reanalysis data, and can be applied for both post-processing and in real time. Finally, the new approach is using anomalies over several height layers, and TEAs over larger area, and hence the detection and monitoring results are not sensitive to details such as exact latitude or pressure level. Potential further refinements of the thresholds for our metrics will be determined from recently started work on multiple SSW events, using longer-term data over the recent decades.*"

**Specific comments**

**Point 1:** *Do the authors report somewhere that bending angle and density are given in normalized units? This is apparent, but the reader would benefit from a definitive statement in the manuscript. As well, please state how the normalization is performed (normalized with respect to what?).*

**Response 1:** We have in fact shown the equations of how to calculate anomalies from (1) to (3) in our Table 1, which is used to concisely summarize our methodology of the full approach. To make this clearer, we have added the following statements from lines 20 to 22 in page 6 in the RM:

*"Temperature anomalies are calculated as absolute values, while density and bending angle anomalies are calculated as relative (percentage) values, by dividing the absolute-value anomaly profiles by the collocated climatological profiles."*

**Point 2:** *Abstract, line 20: recommend "has strong potential."*
**Response 2:** Thanks, we have corrected to "has strong potential" in the RM.

**Point 3:** *Abstract, lines 17 and 22: is it necessary to introduce these metrics – the 3 Mio. km^2 threshold and the MSTA-TEA40 metric – here? I'm not sure that the abstract benefits from either the specificity of the former or the raising of the as-yet undefined metric and abbreviation of the latter. I would recommend removal unless the authors have strong objections.*
**Response 3:** Thanks for this suggestion. In the RM, we have avoided to introduce the metrics in the abstract. We improved the sentence from lines 22 to 23 in the first page in the RM as follows:

*"temperature anomalies over the middle stratosphere exceeding 35 K cover an area more than 10 Mio. km²."*

**Point 4:** *Pg. 8, lines 25-26: I'm not quite sure I follow what's being said here. Is it that the specific definitions the authors have proposed may change as more systematic study is performed?*
**Response 4:** Yes, we try to say that the we can tell whether the SSW event is a strong one or a more minor one from our recorded duration and also strength. However, we agree that we can only know the specific values or thresholds of such determination of SSW events by applying our method to longer-term data records. By then, we can more robustly propose a definition for SSW based on the determination thresholds. To make our statements clearer for now in this initial introduction of the method, we have corrected the corresponding sentences from lines 5 to 6 in page 9 in the RM as follows:

*"However, the specific thresholds for our metrics and indicators for SSW detection, monitoring, and classification can only be determined after the new approach is applied to longer-term data containing multiple events."*

We thank Reviewer 1 for his/her comments again. In addition to the comments and suggestions from the Reviewers, we inform that based on our careful rechecks of the full computation we found that a technical bug remained in our original calculation of the background climatology. It implied that we inadvertently had used the mean climatology of January and February of the year 2017 only, instead of the long-term mean climatology from the full range 2007 to 2017. We have corrected this in the updated RM.

We found that all our basic conclusions are still robust and valid; just the detailed quantifications changed, since the long-term climatology yields a somewhat different (and even more smooth) reference background. Figures 1 to 5 are very similar to the

original ones. The main impact is that the TEAs from the original thresholds are now smaller, while the TEAs temporal characteristics are strongly similar. Therefore, in the RM, we generally selected somewhat lower thresholds for calculating TEAs and for formulating our metrics, to keep their sizes and magnitudes rather similar. The Figures 6 to 7 reflect this and confirm that there was no qualitative change, just limited quantitative change that left the conclusions robust. Please find the RM for further details where we made small adjustments to the text accordingly.

---

## Author Response (AR2)

**Response to Reviewer 3 Comments**

**Overall Comment:** I appreciate the authors providing detailed responses to my previous comments, which have helped me to understand the paper in more depth. The proposed analytical technique has the potential to become a new definition of SSW. However, to confirm this method as a robust and standard definition of SSW, at least several major/minor SSW events must be additionally tested. Based on the responses to Points 2, 4, 16, and 20, this study is inconclusive, where further refinements are required, namely, by investigating a longer-data record. I am concerned, for example, with the response to Point 4, as the selection of TEA parameters, the corresponding height ranges, and their thresholds may be modified depending on each SSW event, and thus could produce ambiguity in defining major, minor, and final warmings.

I would like to suggest that the authors continue the analysis of several additional major/minor SSW events to make this method more concrete and to demonstrate fully the usefulness of the novel GNSS-RO data. However, if the authors are insistent about introducing this method to the SSW community in its present version, I look forward to a follow-up study in which the method is applied to a long-term dataset.

**Response to overall comment:** We thank Reviewer 3 for this overall comment. And yes, we agree that any such new method should be tested with longer-term records. This is why we did refer in this initial intro paper's title to "a new approach demonstrated by the 2009 event", and we confirm we have successor work on-going for the follow-on paper that completes the introduction as a consolidated method, based on long-term RO and ERA5 records with a whole ensemble of SSW events over the recent decades. We found that an "all-in-one single paper" would be really lengthy and we therefore indeed prefer to be "insistent" with this two-papers approach.

Regarding the possible "tuning needs" of various TEA parameter settings, etc., we confirm in view of our preliminary follow-on paper results on a range of SSW events in the RO timeframe 2006 to 2019 that the new approach is robust and that detailed parameter settings can be consolidated without changing the method's basic design. This most up-to-date knowledge we meanwhile have from the follow-on work is in line with what we state in the conclusions of this paper, i.e., that we expect such refinements to work and that we will do such consolidation using long-term records.

**Specific comments to the responses**

Point 1: In addition to the statements cited in the author's response, Butler et al. (2015)

also refers to some useful points about the SSW definition, including the technique's statistical application to SSWs, consistency across observational and modeling studies, and detection of historical SSWs. These suggest that any new method should be tested with long-term records.

The following statement in the concluding section (Butler et al., 2015) provides a major summary of the new SSW definition.

"We believe a new definition should include, at a minimum, guidelines for determining a) the independence of closely timed events; b) the classification of split-type versus displacement-type events; and c) precise distinctions among major, minor, final, and Canadian SSWs."

Have these requirements been met in the current study by considering only the single SSW event from 2009?

**Response to Point 1:** We repeat that we agree that any new such method should be tested with long-term records; see our response to the overall comment above. An all-in-one paper would be lengthy and hence we prefer this two-papers approach. We also agree also that full compliance with these Butler et al. requirements partly needs the results from the longer-term analysis from multiple events; as we ourselves also say already in this manuscript. So overall the requirements are only met by both papers together (hence the first one is introducing the "new approach").

More specifically, regarding these requirements: a) "the independence of closely timed events", this is analyzed more in long-term processing, and we find as part of the follow-on work that our trailing-metric is quite helpful to this end; 2) "the classification of split-type versus displacement-type events", we find split-type events usually show larger anomalies, also this is part of the long-term analysis; 3) "precise distinctions among major, minor, final, and Canadian SSWs", we expect this to be quite robustly possible based on the refined TEA parameter settings and then using our three core indicators (Main-phase duration, Main-phase area, Main-phase strength) as well as optionally the trailing metric.

We have introduced a new small paragraph at the end of Section 3 to explicitly address now the point also that these aspects of the Butler et al. (2015) requirements will (only) be assessed in the follow-on work, but that we are aware of and intend to do so.

**Point 3:** I agree that the zonal wind in a narrow latitude band at a specific height may not be sufficient to define SSW. However, the effects of SSW have been commonly investigated in terms of the behavior of the polar vortex and its effects on stratosphere– troposphere coupling. Therefore, examination of the wind field, including the circulation reversal and the characteristics of planetary waves, is critical and should probably be included in a SSW definition.

**Response to Point 3:** Thanks for the comment. We agree that polar vortex is strongly related to the SSW and that wind data as available can be very useful. However, we also know from many previous studies that the middle stratospheric anomalies, e.g., temperature anomalies, wind anomalies and also geopotential height anomalies are strongly related, and all related to the state and dynamics of the polar vortex. The polar vortex strength is a key factor for stratospheric downward effects on tropospheric circulations and is related to the strength of both temperature and wind anomalies.

Therefore, we see our approach with its temperature anomaly-related ansatz one good method that also from the data (availability) side has advantages over wind field use. Moreover, since ERA5 data also have wind field information, we may as part of the long-term analysis also analyze the relationship between anomalies in thermodynamic variables as used in our method and the polar vortex changes and wind reversals.

**Points 4 and 16:** I understand that the thresholds of TEAs have yet to be confirmed, and they should be adjusted by repeating the analysis for a long-term dataset. This study utilizes various parameters obtained from GNSS-RO measurements over a wide height range, which are summarized as a total of 12 values in Table 1. I am not fully convinced by the selection of only five of the 12 parameters. Are the other parameters useless?

**Response to Points 4 and 16:** In fact, Table 1 shows the methodology flow and that is why we list there quite a number of parameters. (1) to (3) shows how we calculated anomaly profiles for temperature, density and bending angle. Based on (1) to (3), we then have our five TEA values shown from (4) to (8). Then based on these five TEA values, we have our primary, secondary and trailing metrics ((9) to (11)) for SSW monitoring. (12) to (13) are temperature only metrics for readers who prefer only temperature for SSW detection. Based on metrics (9) to (11), we then offer three overall metrics shown in (14) to (16) as the main metrics for SSW definition and classification. Hence, all parameters along the sequence are useful, and the final ones are those metrics that are mainly used to track the SSWs characteristics in the long-term records.

**Points 5 and 12:** Unfortunately, I do not fully understand the response. Provided GNSS-RO data is assimilated into the ECMWF, they both naturally provide nearly the same results. Do the authors suggest employing ECMWF for application of this method to the larger re-analysis data from the past decades, which is much longer than the GNSS-RO records? **Response to Points 5 and 12:** Yes, formally the method can be applied to any reliable gridded data and we used in this paper the RO and ECMWF data to demonstrate this. And specifically to the question: yes, as we briefly describe in the paper's conclusions, in our follow-on paper work we use both the GNSS-RO data and the longer-period (ERA5) re-analysis data, where the latter indeed strongly back-extend the record.

**Point 6:** I am interested in the retrieval technique when the model atmosphere is not used. Please show the citation for the non-optimized method.

**Response to Points 6:** Thanks for this comment. Based on Reviewer's suggestions in previous first review, we already used the non-optimized bending angle in our revised manuscript. Therefore, the model atmosphere is not influencing now anymore.

We again thank Reviewer 3 for his/her valuable comments.